# Endothelial Dysfunction in Obesity-Induced Inflammation: Molecular Mechanisms and Clinical Implications

**DOI:** 10.3390/biom10020291

**Published:** 2020-02-13

**Authors:** Ibrahim Kalle Kwaifa, Hasnah Bahari, Yoke Keong Yong, Sabariah Md Noor

**Affiliations:** 1Department of Pathology, Faculty of Medicine and Health Sciences, Universiti Putra Malaysia (UPM), Selangor 43400, Malaysia; nanamarye2009@gmail.com; 2Department of Haematology, School of Medical Laboratory Sciences, College of Health Sciences, Usmanu Danfodiyo University (UDU), Sokoto, North-Western 2346, Nigeria; 3Department of Human Anatomy, Faculty of Medicine and Health Sciences, Universiti Putra Malaysia (UPM), Selangor 43400, Malaysia; haba@upm.edu.my (H.B.); yoke_keong@upm.edu.my (Y.K.Y.)

**Keywords:** obesity, adipose tissue, inflammation, atherosclerosis, endothelial dysfunction

## Abstract

Obesity is characterized by the excessive deposition of fat that may interfere with the normal metabolic process of the body. It is a chronic condition associated with various metabolic syndromes, whose prevalence is grossly increasing, and affects both children and adults. Accumulation of excessive macronutrients on the adipose tissues promotes the secretion and release of inflammatory mediators, including interleukin-6 (IL-6), interleukin 1β, tumor necrotic factor-α (TNF-α), leptin, and stimulation of monocyte chemoattractant protein-1 (MCP-1), which subsequently reduce the production of adiponectin thereby initiating a proinflammatory state. During obesity, adipose tissue synthesizes and releases a large number of hormones and cytokines that alter the metabolic processes, with a profound influence on endothelial dysfunction, a situation associated with the formation of atherosclerotic plaque. Endothelial cells respond to inflammation and stimulation of MCP-1, which is described as the activation of adhesion molecules leading to proliferation and transmigration of leukocytes, which facilitates their increase in atherogenic and thromboembolic potentials. Endothelial dysfunction forms the cornerstone of this discussion, as it has been considered as the initiator in the progression of cardiovascular diseases in obesity. Overexpression of proinflammatory cytokines with subsequent reduction of anti-inflammatory markers in obesity, is considered to be the link between obesity-induced inflammation and endothelial dysfunction. Inhibition of inflammatory mechanisms and management and control of obesity can assist in reducing the risks associated with cardiovascular complications.

## 1. Introduction

Incidences of obesity are increasing exponentially, and this has contributed effectively to the increasing prevalence of various pathological conditions of obesity-related metabolic disorders. Currently, about 1.9 billion people are estimated to be obese or overweight globally, with an approximated figure of 50 million affected children suggested to be within 5 years of age [1]. Obesity is a worldwide problem affecting both developed and poor-resource countries [2], described as the excessive deposition of body fat, reported by the body mass index (BMI) and calculated as weight (kg) divided by height (meters) squared. Individuals with BMI values greater than 19 and less than 25 kg/m^2^ are considered healthy, while people with a BMI equal or greater than 25 kg/m^2^ are overweight and those with a BMI equal or greater than 30 kg/m^2^ are considered obese [2]. Although the prevalence of obesity in developed countries is reported to have slowed in the past few years, the rate in developing countries continues to increase and might have even tripled in some developing countries over the past few years [3,4]. This is strongly attributed to growing changes in lifestyle, reduced physical activity, and availability of modified junk foods [4,5]. Obesity is described to result from increased energy intake with subsequent reduced physical activities [2]. It is a metabolic condition of chronic low-grade inflammation, associated with high levels of inflammatory markers, such as CRP, IL-6, and TNF-α [2,6]. In this, the low-grade chronic inflammation is characterized by a variety of chronic diseases, including cardiovascular disease, diabetes, hypertension, nonalcoholic fatty liver disease [2], hypercholesterolemia, asthma, arthritis, some cancers, and general poor health condition, and hence represents a significant burden on the global healthcare system [5]. The global prevalence of obesity increases the potential risk factors for developing chronic metabolic syndromes [7]. Significant increase in bodyweight is highly attributed to several metabolic complications, such as cardiovascular disease, pulmonary complications, diabetes, and some certain cancers [8,9], leading to diminished life expectancy [2,6]. Furthermore, overweight or obese pregnant women have increased risks of developing offspring with obesity and its other related metabolic complications in future life, which justifies a transgenerational cycle of obesity [5]. Obesity is linked with alterations in immunity attributed to elevated levels of circulating proinflammatory cytokines [10]. Obesity usually results from an imbalance between energy intake and utilization, leading to a significant difference in system energy balance [11]. In recent years, research for a molecular basis linking the pathophysiology of obesity with inflammation has shown a close correlation between the accumulation of excessive macronutrients and stimulation of the system immunological responses from various tissue related to system homeostasis [12]. Previous reports also have shown that inflammation could result from obesity, which is attributed to overexpression of MCP-1, and consequently, generating defective insulin secretion, insulin resistance, and interfering with other processes of energy homeostasis as well [11]. The resultant failure of adipose tissue to adequately overcome the intake fat, leads to the excessive accumulation of fat to vital organs of the body, including liver, kidney, and heart, resulting in metabolic syndrome. Adipose tissue hypertrophy due to physical means also promotes organelles and cell ruptures resulting in an inflammatory reaction. In addition, oxidative stress associated with excessive intake of fat and other macronutrients devoid of antioxidant-rich food results in inflammation attributed to obesity [13]. Inflammation functions as a defensive mechanism for tissue responses to injury or tissue damage through remodeling, reducing, or even destroying the contributing agents or removing the entire tissue, in the process known as apoptosis. Obesity-induced inflammation is a multifaceted condition affecting many organs, including skeletal muscle, adipose tissue, liver, brain, heart, and pancreas [14,15]. Endothelial dysfunction (ED) results from an imbalance in the production of vasodilatory agents, such as nitric oxide (NO), endothelial-derived hyperpolarizing factors (EDHF), prostacyclin (PGI2), and vasoconstricting agents, including prostaglandin (PGH2), endothelin-1 (ET-1), and angiotensin-II (Ang-II). Under normal physiology, the balanced release of contracting and endothelial-derived relaxing factors is maintained [6]. Alterations in this balance predispose the vascular endothelium towards prothrombotic and proatherogenic states, resulting in platelet activation, leukocyte adherence, vasoconstriction, pro-oxidation, mitogenesis, vascular inflammation, impaired coagulation, atherosclerosis, and thrombosis with subsequent cardiovascular diseases [6]. During obesity, this delicate balance is usually disrupted, promoting the development and further progression to vascular endothelial dysfunction with subsequent damage to some vital organs [5,6]. Thorough knowledge and understanding of the basic mechanisms associated with obesity-induced inflammation are required so that efficient therapeutic programmed or prophylactic strategies to prevent these preventable but life-threatening conditions can be developed. This review summarizes the pathophysiology of the endothelium, and the molecular mechanisms through which adipose tissue secretions in obesity influence endothelial dysfunction. The functions and contributions of MCP-1 to endothelial dysfunction in obesity are highlighted in detail. The mechanisms linking obesity, inflammation, and endothelial dysfunction are also discussed.

## 2. Obesity and Metabolic Disorders

In 1988, “syndrome X” was discovered by Reaven and named as the “metabolic syndrome”. Previous reports indicated that several factors have been linked to the progression of cardiovascular diseases; these include elevated triglycerides and low-density lipoprotein (LDL) cholesterol levels, glucose intolerance, hyperinsulinemia, an elevated level of non-esterified fatty acids (NEFAs), reduced high-density lipoprotein (HDL) levels, and hypertension [16]. Metabolic syndrome represents a collection of conditions that arise together as predisposing factors to the progression of kidney, heart, and liver diseases. Metabolic syndrome associated with obesity is a fast increasing epidemic worldwide, and their incidence increases with age, this justified its high prevalence in people over 50 years compared to other age groups [17]. The National Cholesterol Education Program’s Adult Treatment Panel III report (ATP III) described six mechanisms of metabolic syndrome contributing to the progression of cardiovascular diseases; hyperlipidemia, high blood pressure (HBP), insulin resistance (IR), obesity, atherogenesis, as well as proinflammatory and prothrombotic state, on which obesity became the “cornerstone” of the syndromes [18,19]. Obesity is considered to be the core figure contributing to the increased morbidity and mortality resulting from metabolic syndrome, and these are attributed to various factors including ineffective activities of the renin-angiotensin system, elevated proinflammatory cytokines activities, vasoconstriction from increased activities of sympathetic nervous system, deregulation of adipokines synthesis and secretion, alteration in the level of circulating insulin, and bioavailability of insulin-like growth factor-1 (IGF-1) [6]. Additionally, vascular endothelial integrity-related factors including vascular endothelial growth factor-1 (VEGF-1), synthesis of vascular reactive oxidative species (ROS), plasma plasminogen activator inhibitor-1 (PAI-1) thrombotic-mediated tendency, hyperuricemia, triglyceride, and oxidation-prone LDL-c levels are found to be elevated during obesity and its related metabolic syndrome [6,9]. 

### 2.1. Impact of Obesity on Adipose Tissue

Mammalian adipose tissues are described as white and brown. Most of the mammalian adipose tissue is white, which is described to be the area of energy storage [20]. However, brown adipose tissue is usually elevated abundantly in neonates functioning effectively to regulate body temperature and energy utilization during non-shivering thermogenesis promoted by the uncoupling protein-1 (UCP1) [8]. In human adults, the extent of brown adipose tissue in the body correlates to body mass index (BMI), signifying the contribution of brown adipose tissue in metabolic processes [21]. Adipokines produced by adipose tissue actively participate in controlling the physiologic and pathologic processes such as adipocyte differentiation, lipid and glucose metabolisms, cardiovascular and neuroendocrine functions, immunity, and inflammation processes. Adipose tissue synthesizes and releases various proinflammatory and anti-inflammatory factors, including adipokines, cytokines, and chemokines, which are mostly elevated in obesity [21]. Majorly, inflammation of the adipose tissue is characterized by the infiltration of the immune cells including macrophages, neutrophils, and B- and T-lymphocytes. Macrophages infiltration of the adipose tissues is the major contributor of inflammation associated with endothelial dysfunction [22]. The report has indicated that Macrophage-specific genes, including MCP-1, macrophage inflammatory protein-1α (MIP-1α), CD68, CD11b, and F4/80, were shown to be up-regulated in the adipose tissues of obese mice. Immunohistopathological analysis of adipose tissues revealed a significant relationship between the percentage of F4/80 expression, adipocyte size, and body mass [22].

### 2.2. Obesity, Inflammation, and Related Metabolic Syndrome: The Linking Mechanisms

Normally, the adipose tissue is mainly the collection of preadipocytes and adipocytes, with few leukocytes. However, during obesity, the composition and functions of adipose tissue are altered. Once the adipose tissue’s storage capacity is saturated with fat, the incoming fat may be stored intra or extravascularly in other vital organs, including the skeletal muscle, heart, kidney, and liver, leading to localized IR to these vital organs. Excessive energy consumption without utilization causes adipocytes to undergo hypertrophy (increased adipocyte volume), which can result in further complications such as hypoxia, adipocyte necrosis, chemokine secretion, and irregular fatty acid flux. Adipocytes hypertrophy disturbs the balance of adipose tissue-derived cytokines and adipokines, leading to a proinflammatory state with subsequent adipocytes cellular dysfunction, which inhibits adiponectin mRNA expression and stimulates endoplasmic reticulum stress [22]. This is also associated with increased secretion of proinflammatory cytokines and MCP-1, causing a typical infiltration of activated macrophages that further promotes the inflammatory process [23]. The level of circulating leptin concentration is elevated with the subsequent reduction in the concentration of adiponectin [24]. Inflammation is associated with metabolic abnormalities and vascular dysfunction, indeed, elevated levels of circulating inflammatory markers are observed in patients with obesity [5]. Overexpression of proinflammatory cytokines during obesity is considered to be the link between obesity and inflammation (Figure 1) [15].

## 3. Endothelium and Endothelial Cells Functions

The endothelium is the natural inner lining of the vessels, known to regulate and coordinate the vascular and organs integrities [6]. The layer is arranged: tunica intima, consists of endothelial cells; tunica media, forms the vascular smooth muscle cells (VSMC); and tunica adventitia, elastic lamina comprised of terminal nerves fibers around connective tissues. Normally, endothelium functions to regulate vascular homeostasis through coordination of blood flow, distribution of nutrients, hormones, and other macronutrients, and migration and proliferation of VSMC, which controls coagulation and fibrinolysis activities, reduces vascular tone and regulates cellular and vascular adhesions, inhibits leukocyte adhesions, and modulates inflammatory activities and angiogenesis (Figure 2) [5]. Vascular endothelium also inhibits platelet adhesions and aggregations through the production of prostacyclin (PGI2) and nitric oxide (NO) by ecto-adenosine diphosphate (ADP)-ases, CD39, CD73 exposures, and prostaglandin E2 promoted by glycosaminoglycans of the endothelial cell surface. Endothelium controls adhesion and activities of the coagulation factors through the interactions of thrombomodulin (TM), antithrombin III (ATH III), tissue factor (TF), and tissue factor pathway inhibitor (TFPI), and it regulates tissue-type plasminogen activator-1 (tPA-1), urokinase plasminogen activator-1 (uPA-1), and their inhibitor; PAI-1 [6,25]. To perform these tasks, the endothelium secrets endothelial cells moieties (through its paracrine and autocrine activities) and other regulatory mediators such as NO, prostanoids, endothelin-1 (ET-1), angiotensin II (Ang II), t-PA, PAI-1, von Willebrand factor (vWF), adhesion molecules, and cytokines, which are responsible for various stimuli. The bioactive substances secreted by endothelial cells also control and maintain the structures and functions of intact blood vessels by balancing between inflammatory and anti-inflammatory factors, proliferative and antiproliferative agents of VSMCs, oxidative and anti-oxidative agents, dilations and contractions of vasculature, blood coagulation cascade and fibrinolytic system [14].

The anticoagulant activities of the endothelium are attributed to the synthesis of TM and heparin sulfate proteoglycan, and the secretion of TFPI. Thrombomodulin binds to thrombin, which stimulates protein C with eventual inhibition of FVIII and FV activities. The heparin-like molecule serves a cofactor to ATIII, while TFPI reduces FXa and TF-FVIIa complex activities. The tPA of the endothelium activates the fibrinolytic system, while prostacyclin (PGI2) and nitric oxide are synthesized to regulate and maintain the antiplatelet activities of the endothelium [26]. Nitric oxide is a gas constantly produced in the endothelium from the L-arginine by the synthesized endothelial nitric oxide synthase (eNOS). Expression of eNOS following Ca^2+^ (calmodulin) complex produces NO, promoting L-arginine and other cofactors such as NADPH and tetrahydrobiopterin together as substrate. The VSMC absorbs NO, which forms a complex with a heme-iron group of guanylate cyclase to form a cyclic guanosine monophosphate (cGMP), which activates a cGMP-dependent protein kinase leading to elevated Ca^2+^ of the cytosol in VSMC; this suppresses vasoconstriction and initiates vasodilation. Nitric oxide is a potent endothelium-derived vasodilator functioning to have antiplatelet, antiproliferative, anti-inflammatory and permeability decreasing properties. Furthermore, NO also combines with endothelium-derived hyperpolarizing factor (EDHF) and Kruppel-like factor 2 to facilitate arterial vasodilation [27]. Nitric oxide inhibits the synthesis of Vascular Cell Adhesion Molecule-1 (VCAM-1) and MCP-1 activities, resulting in the decreased expression of nuclear factor κB (NF-κB) [20]. Ischemia and shear stress stimuli among others were also reported to trigger the release of NO, particularly the synthesis of Weibel-Palade bodies (WPB), when activated commonly by histamine or thrombin. These granules can be promoted by other facilitators of the NF-κB pathway including CD40L and Toll-like receptors (TLR), which contribute to NF-κB signaling molecule in achieving degranulation [28,29,30]. These result in the production of ultra-large vWF (ULVWF) multimers, which bind to platelets to form a complex with GPIba or GPIIb. These multimers are later destroyed by metalloproteinases ADAMTS13 at endothelial lumen, thereby initiating a feedback mechanism to inhibit platelets adhesion and aggregation [14].

### 3.1. Endothelial Dysfunction

Under normal physiological conditions, the secretion and release of endothelial-derived vasodilating and vasoconstricting factors are maintained and balanced. However, in obesity-induced atherosclerosis, this delicate balance is distorted, further promoting the progression of vascular endothelial dysfunction and end-organs damage [31]. Endothelial dysfunctions are usually characterized by the imbalance in the secretion and release of vasoconstriction and vasodilation agents, thereby predisposing the vascular endothelial towards prothrombotic and proatherogenic effects. Defective endothelial physiological properties result in leukocyte adhesion, activation of platelets, pro-oxidation of mitogens, impaired PGI_2,_ coagulation and nitric oxide productions, decreased synthesis of EDHF, and vasoconstriction factors including Ang II and prostaglandin (PGH2), atherosclerosis, and thrombosis [31]. The fundamental mechanisms attributed to the progression of endothelial dysfunction in obesity are numerous including elevated levels of LDL and triglycerides, increased oxidative stress radicals, elevated levels of inflammatory factors, and imbalanced hemodynamic activities. The major participating agents of endothelial dysfunction in obesity include IR, oxidized form of low-density lipoprotein (oxLDL), adipose tissues related inflammation, and decreased NO bioavailability [31]. Other contributing factors of endothelial activation and deregulation include decreased tetrahydrobiopterin (BH4) bioavailability and elevated eNOS uncoupling, elevated production of ROS and arginase, elevated glycation, and synthesis of the receptor for advanced glycation end products (RAGE). In addition to these are; reduced NO bioavailability, increased asymmetric dimethylarginine, activation of NFκB, inhibition of Kruppel-like Factor 2 [32], and phenotypic alterations in perivascular adipose tissue resulting in mild inflammation and elevated leptin with subsequent reduction of adiponectin secretions [33,34]. Defective biosynthesis of NO and facilitated secretion of ROS by uncoupled eNOS, nicotinamide adenine dinucleotide phosphate (NADPH) oxidases, xanthine oxidase, lipoxygenase, cyclooxygenase, microsomal P-450 enzymes, and pro-oxidant heme molecules are considered as the molecular mechanisms for endothelial dysfunction [6,35]. Additionally, the secretion of angiotensinogen of the renin-angiotensin-aldosterone system (RAAS) by the dysfunctional adipocytes could lead to its overexpression in the RAAS system, and subsequent elevated production of ROS [36]. Three isoforms of NOS are described; the constitutive comprising the neuronal NOS (nNOS or NOS-1) and endothelial NOS (eNOS or NOS-3), which are the major contributors of the circulating nitric oxide together with the inducible NOS (iNOS or NOS-2) [37,38]. Variant eNOS genes mostly promote endothelial dysfunction by attenuating NO production [6]. Stimulation of NF-κB signaling molecules of endothelial cells results in proadhesive and procoagulant phenotypes with associated defective barrier functions [16]. Furthermore, NF-κB target genes present in endothelial cells are adhesion molecules, including intercellular adhesion molecule-1 (ICAM-1), VCAM-1, and E-selectin, which intervene with inflammatory cytokines of the vascular wall to promote extravasations and subsequent endothelial dysfunction [14,39].

The term inflammation is associated with edema, transmigration, infiltration of leukocytes, and blood vessel and connective tissue proliferations. Inflammation has been considered as the initial stage of vascular dysfunction, progressing to vascular disease related to obesity [5,15]. Factors that promote atherogenesis in obesity include ox-LDL, Ang II, and hyperglycemia, which facilitate the activities of NF-κB and MAPKs in endothelium by leading to the stimulation of proinflammatory cytokines, chemokines, increased synthesis of ICAM-1 and VCAM-1, activation of iNOS, growth factors, and other enzymes [40]. These responses stimulate the production of interleukins, including IL-1β and IL-18, which facilitate the progression of inflammation through activation of proinflammatory signaling complexes of the inflammasomes and oligomerization domain-like receptor family pyrin domain containing 3 (NLRP3) [41]. Inflammation can be screened through the examination of inflammatory markers, popularly known as high sensitivity C-reactive protein (hs-CRP) and the inflammatory score derived from the proinflammatory cytokines, osteopontin, chemokines (C-C motif) ligand 2 (CCL2), CCL5, chemokines (C-C motif) ligand 5, cyclooxygenases (COX), connective tissues growth factors (CTGF), fractalkines (CX3CL1), iNOS, ICAM-1, NFκB and transforming growth factor β (TGFβ), TLR, and anti-inflammatory adiponectin [42,43,44]. Other related biomarkers include: growth differentiation factor-15 (GDF15), myeloid-related proteins 8/14, pentraxin 3, and lectin-like oxidized low-density lipoprotein receptor-1 (LOX-1), which were identified as surrogate markers of atherosclerosis associated with cardiovascular diseases during obesity [45,46], while galectin-3 was described as a potential biomarker of vascular remodeling and endothelial dysfunctions attributed to inflammation in obesity [47].

### 3.2. Coagulation System and Endothelial Dysfunction

Under normal physiology, adipose tissue is mainly comprised of adipocytes, preadipocytes, and some leukocytes. Adipokines synthesized by adipose tissue contribute significantly in regulating the physiologic and pathologic processes, particularly the adipocytes differentiation, immunity and inflammation processes, and lipid and glucose metabolisms. Obesity is characterized by an alteration in the normal composition of adipose tissue. However, excessive deposition of fat in obesity leads to adipose tissue dysfunction resulting in adipocytes hypertrophy. Adipocytes hypertrophy alters the balance of adipose tissue-derived cytokines, hemostasis, and adipokines, leading to proinflammatory and prothrombotic states [22]. These trigger the increased secretion of proinflammatory cytokines, prothrombotic markers, and a typical infiltration of endothelium by activated macrophages, further promoting inflammatory process [23]. Indeed, inflammation is associated with elevated levels of circulating inflammatory cytokines and prothrombotic markers observed in patients with obesity-related endothelial dysfunction [5]. The predominant macrophages found in adipose tissue also produce TF, which combines with the elevated liver secretion of FVII and FVIII to promote the possibility of coagulation abnormalities. Furthermore, adipose tissue in obesity secretes decreased levels of adiponectin, thereby facilitating the susceptibility of platelets aggregation with the subsequent increased PAI-1 production, which further inhibits fibrinolysis. All of these conditions are reported to contribute significantly to the progression of the prothrombotic state found in obesity and its other related syndromes [27]. Normally, active endothelium serves as an antithrombotic surface, thereby regulating the activities of the coagulation cascade [5]. Endothelial cells (ECs) are the major contributor of the components responsible for proinflammatory, procoagulant, and antifibrinolytic activities, and counter effectors with the anticoagulant, anti-inflammatory, and profibrinolytic properties. These components are vital for the regulation of physiologic balance associated with control and coordination of the vascular hemostatic system (Figure 2) [5]. However, inflammation has been characterized by the imbalance between proinflammatory and procoagulant, and anti-inflammatory and anticoagulant activities of the endothelium, leading to disturbance of the hemostatic system. Once stimulated, ECs produce procoagulant or antifibrinolytic components and the subsequent reduction in the secretion of anticoagulant and profibrinolytic components with further aggravation of endothelial dysfunction (Figure 3) [48].

Formation of thrombus is regulated and takes place by the majority components of the hemostatic system such as vascular ECs, platelets and plasma, physiologic anticoagulant pathways (PAP), coagulation cascade, and fibrinolytic activities. However, early inflammatory responses are initiated by inflammatory mediators; particularly the proinflammatory cytokines that have significant effects on the hemostatic system, as well as mediating the procoagulant activities on the dysfunctional endothelium [49,50]. Vascular inflammatory mediators disturb the hemostatic system using various mechanisms, such as endothelial cells dysfunctions, activation of platelets, the TF-mediated stimulation of the plasma coagulation system, and suppressed functions of PAP and impaired fibrinolytic activities (Figure 4) [50].

### 3.3. Endothelial Dysfunction during Vascular Aging and Cellular Senescence

Atherosclerosis leading to cardiovascular complications, including myocardial infarction, stroke, and ischemic heart failure, is considered to be the main cause of death in the Western world [51]. Atherosclerosis is well known to be associated with diabetes, LDL, cholesterol, smoking, and hypertension. Recent reports have indicated that aging is also one of the important risk factors for atherosclerosis and continues as an independent contributor when all other known factors are excluded [51]. Atherosclerosis-induced endothelial dysfunction is; therefore, a disease of both organismal aging and cellular senescence. During advanced age, ECs become flattened and enlarged with an increasingly polypoid nucleus and all features identified with cellular senescence [52]. These alterations are associated with angiogenesis, proliferation, and cell migration and modulation in cytoskeleton integrity. Vascular endothelial senescent shows reduced endothelial NO production with the elevated endothelin-1 release, decreased expression of VCAM-1 and ICAM-1, raised activation of NF-kB, and enhanced vulnerability to apoptosis [52]. Thus, senescent EC is attributed with the loss of EC activities and a subsequent shift toward a proinflammatory and proapoptotic state, predicted to promote monocyte migration into the vessel wall, leading to endothelial dysfunction [51]. Naturally, vascular endothelial aging is associated with intimal and medial thickening with a gradual loss of vascular elasticity, resulting in vascular endothelial stiffness [53]. Natural and premature aged cells shared various common characteristics, including alterations in cell proliferative potentials, changes in markers of cell senescence, increased apoptosis, increased DNA damage with tremendous telomere shortening and dysfunction, decreased medial VSMC, elevated collagen deposition, and fracture of the elastin lamellae leading to vessel dilation and expanded lumen size. These are promoted by age-associated elevated glycated proteins, matrix metalloproteinase enzyme activities, and trophic stimuli, including angiotensin II signaling and impaired vascular endothelial elasticity progressing to hypertension [51]. Cell senescence is described as the irreversible loss of the ability of the cells to divide and has been categorized into replicative senescence and stress-induced premature senescence (SIPS). While replicative senescence progresses with age and is identified by shortened telomeres at chromosomal ends, which then promotes a DNA damage response (DDR), the SIPS on the other hand is triggered by external stimuli, such as radiation and oxidizing agents, which trigger the intracellular senescence cascade prematurely, resulting in vascular endothelial dysfunction [51]. Both normal vascular aging and atherosclerosis are associated with cellular senescence. Cellular senescence impairs cell proliferation, resulting in irreversible growth arrest and impairs survival due to an accumulation of nuclear and mitochondrial DNA damage, increased ROS, and a proinflammatory state. Vascular aging and cellular senescence are also associated with increased expression of proinflammatory cytokines and adhesion molecules, which further promote inflammation and affect the synthesis and maintenance of extracellular matrix proteins. Aging can be identified by both structural changes and by many senescence-associated biomarkers [51].

### 3.4. Endothelial Dysfunction and Epigenetic Modifications

Epigenetic modifications such as DNA methylation and histone acetylation are described as post-replication changes in chromatin without any alteration in the basal nucleotide code [51]. These mechanisms are essential in controlling gene activation and silencing, and they are connected to various age-related conditions including atherosclerosis [54]. Epigenetic modifications are therefore utilized as biomarkers of cellular senescence in the vascular endothelial. Previous studies have recognized certain changes in the expression of methyltransferases, indicating that alterations in methyltransferase expression are linked with hypomethylation of hypermethylated genomic regions that occur within the genes known to participate in lipid metabolism, proliferation, and apoptosis. Atherogenic lipoproteins can induce DNA methylation and histone deacetylation [51]. These findings proposed a potential link between epigenetic modification and atherosclerosis. Histone acetylation and deacetylation are known to contribute significantly to the progression of atherosclerosis during aging through which inflammation, VSMC proliferation, and ECM composition can be modulated. Furthermore, other studies have indicated that mitochondrial gene p66 (Shc), a known longevity gene, promotes hypermethylation and histone acetylation that results in age-related enhancement of p66Shc production and its subsequent activation [55].

### 3.5. Endothelial Dysfunction Influenced by Reactive Oxygen Species (ROS)

Oxidative stress is described as a state of an imbalance between the bodies pro-oxidant and antioxidant systems, leading to platelet aggregation, thrombus formation, and subsequent endothelial dysfunction [56]. It also alters pancreatic insulin secretion and glucose metabolisms in muscle and adipose tissue. Obesity and its related metabolic abnormalities are associated with increased oxidative stress radicals with elevated expression of NEFAs, TNFa, CRP, IL-6, TNF-α, and LDL cholesterol. ROS also serves as a precursor to Ox-LDL formation, essential to the progression of atherosclerotic lesions. Glucose auto-oxidation in hyperglycemia and protein glycation also contribute significantly to free radical formation [57]. In contrast, HDL particles have both anti-inflammatory and antioxidative activities attributed to its ability to inhibit obesity-associated dyslipidemia. Other antioxidant defence mechanisms that are reduced in obesity include decreased erythrocyte glutathione and glutathione peroxides [58]. In adipose tissues, the secretions of functional adipocytes are tightly controlled by inflammatory and metabolic signals. Adiponectin and leptin are prime hormones adiposity signals secreted from non-obese and noninflamed adipocytes. While leptin acts primarily in the hypothalamus to control food intake and energy consumption, adiponectin secretion, on the other hand, is related to a reduced total body fat mass, which promotes whole-body insulin sensitivity [58]. The secretion of other factors including IL-6, IL-8, chemerin, MCP-1, PAI-1, RANTES, resistin, retinol-binding protein 4 (RBP4), TNF-*α*, or visfatin are significantly raised in adipocytes [59]. Some of these factors induce peripheral complications, vascular endothelial cell dysfunction, atherosclerosis, or cell-mediated inflammatory processes [60,61]. Adipocytes hypertrophy enhances lipid peroxidation with the subsequent generation of ROS, particularly; the generation of reactive aldehyde species (4-hydroxyalkenals) from polyunsaturated fatty acids is elevated. The *ω*-3- and *ω*-6 PUFAs are the reactive aldehydes 4-hydroxy-2*E*-hexenal (4-HHE), 4-hydroxy- 2*E*nonenal (4-HNE), and 4-hydroxy-2*E*, 6*Z*-dodecadienal (4-HDDE), which are the main peroxidation products from which 4-HNE has been identified [62]. This is generated from a series of non-enzymatic peroxidation reactions of 15-hydroperoxy- 5*Z*,8*Z*,11*Z*,3*E*-eicosatetraenoic (15-HpETE) and 13-hydroperoxy- 9*Z*,11*E*-octadecadienoic acid (13-HpODE), 15-lipoxygenase (15-LO)-mediated transformation of arachidonic acid (AA), and linoleic acid. Other compounds include the 12-hydroperoxy- 5*Z*,8*Z*,10*E*, 14-*Z*eicosatetraenoic acid (12-HpETE), the 12- lipoxygenase (12-LO) metabolite of AA, which are also transformed to 4-HDDE. The non-enzymatic peroxidation of *ω*-3 PUFAs, such as *α*-linolenic acid, eicosatetraenoic acid, and docosahexaenoic acid were also reported to generate 4-HHE [63]. ROS are generated from the cells as byproducts of oxidative phosphorylation. ROS promote mitochondrial DNA damage and dysfunction by the addition of double bonds to or removing the hydrogen atom from DNA bases. ROS are increased by elevated levels of oxidized lipoproteins in atherosclerosis, with the most common form being reactive hydroxyl free radicals (OH). Oxidative DNA damage occurs in mitochondrial DNA, in both telomeric and nontelomeric regions [64]. Indeed, many free radicals such as hydrogen peroxide (H_2_O_2_), oxide radical (O^-^), superoxide radical (SO^-^), and the level of 8-oxo-deoxyguanosine (8-oxodG); an oxidized form of guanine contributing to oxidative DNA damage are elevated in VSMCs and macrophages [65]. Improved antioxidant therapy targeted at reducing lipid peroxidation and neutralization of these radicals may eliminate inflammation and improve adipose tissue functions in obesity and its other related metabolic syndrome [66].

### 3.6. Endothelial Dysfunction and Vascular Calcification

Vascular calcification (VC) is one of the mechanisms that influences vascular remodeling due to dedifferentiation of vascular smooth muscle cells (VSMC), alterations in elastin, collagen, and endothelial dysfunction [67]. VC increases the chances of cardiovascular mortality and morbidity, especially in individuals with obesity, type 2 diabetes mellitus (T2DM), and chronic kidney disease [68]. Previously, VC has been described as a degenerative disease or as a consequence of aging [69]. However, recent studies have reported that the mechanism of vascular calcification is a multifactorial tightly controlled process similar to osteochondrogenesis, and it is associated with inflammation, dysregulated metabolism, osteogenesis, and advanced atherosclerosis [70]. The sclerotic process in the vascular endothelium is considered to start from the fatty streaks present as early as childhood, progressing to atherosclerotic lesions, which would be found in many young adults, and may gradually advance to calcified lesions and plaques [71]. Several studies have identified the correlation between ROS generation, particularly H_2_O_2_, and the progression of vascular calcification. An elevated level of ROS triggers MMP (matrix metalloproteinase) activity and alteration in collagen and elastin deposition [72]. MMPs are a family of Zn^2+^ and Ca^2+^ dependent endopeptidases expressed by many different cell types. Vascular endothelial cells can also produce MMPs, which plays a vital role in vascular calcification by degrading the extracellular matrix (ECM) component [69].

### 3.7. Impact of Human Gut Microbiota on Vascular Endothelim

Atherosclerosis is the most common risk factor for CVD, characterized by excessive deposition of cholesterol and recruitment of macrophages into vascular walls, contributing to the development and formation of atherosclerotic plaques [73]. Gut microbiota is the collection of bacteria that inhabit the gastrointestinal tract [74]. They are mainly composed of five phyla, namely Bacteroidetes, Firmicutes, Actinobacteria, Proteobacteria, and Cerrucomicrobia, in which Bacteroidetes and Firmicutes are found more abundant in obese individuals [73,74]. Several studies have indicated that gut microbiota plays a contributing role in atherosclerosis through modulating inflammation and the secretion of microbial metabolites [75]. Recent studies have also shown the influence of gut dysbiosis in the development and progression of atherosclerosis and subsequently CVD [76]. Furthermore, the *Akkermansia muciniphila* was identified to promote gut barrier functions and have attenuating effects against atherosclerosis [77]. Additionally, the human gut microbiota is associated with obesity, and some members of the gut microbiota found to be present in the feces of atherosclerotic patients, are also present in their plaques [78,79]. Human gut microbiota derives energy from dietary fiber through fermentation and produces short-chain fatty acids (SCFAs) to influence host lipid energy metabolism [80]. Diet plays a significant role in modulating microbial diversity and reports have indicated that a high-fat diet is associated with obesity, whereas a fiber-rich diet has the potential for reducing the risk of obesity [81,82]. Gut microbiota play a critical role in hemostasis for maintaining human health, with gut dysbiosis contributing to the development and progression of various diseases including CVD, obesity, T2DM, NAFLD, and even some types of cancer [83].

## 4. MCP-1: A Biochemical Marker Associated with Endothelial Dysfunction

Chemokines represent the family of chemoattractant cytokines categorized according to the number and space occupied by the conserved cysteine residues at the protein N-terminal. They belong specifically to the β family of C–C (cysteine-cysteine), playing a vital role in the recruitment of monocytes, activated neutrophils, lymphocytes, and basophils, while intervening in the IL-1β induction of MCP-1 within the vascular endothelial cells, as well as chemostatic induction of G-protein-coupled through the activation of its receptors [51]. Four sub-families (CXC, CC, CX3C, and C) have been described, with CC (MCP-1) playing the most prominent role during inflammation. MCP-1 is synthesized by several types of cells, including inflammatory and inflammation-mediated cells, monocytic cells, human tubular epithelial cells (TECs), and renal-mediated cells in response to various stimuli [84]. MCP-1 is encoded by chromosome 17q11.2–q21.1 described as MCAF (monocyte chemotactic and activating factor). During obesity, MCP-1 binds to CC chemokine receptor 2 (CCR2) to initiate various monocyte-mediated proinflammatory signals and monocyte chemoattractant activities, facilitating monocytes migration to the subendothelium and combines with ox-LDL to form foam cells, forming a fatty streak and eventual atherosclerotic plaque (Figure 5) [85]. Recent studies indicated that CCR2 is also present in vascular endothelial cells. Activation of CCR2 by MCP-1 was reported to be responsible for the renewal of the vascular endothelium following injury, angiogenesis, and collateral formation. These processes may be essential during inflammatory lesion and tumor metastasis such as atheromatous plaques. Unfortunately, the detailed mechanism by which MCP-1 promotes angiogenesis is still under investigation [86]. Moreover, during endothelial dysfunction, monocytes and their CX3CR1derivatives are recruited by MCP-1 to the site of inflammation promoted by the synthesis of MCP-1; CCL2 and c-Jun N-terminal kinases (JNK1 and JNK2) in adipose tissues and macrophages, respectively. MCP-1 is a major chemoattractant for monocytes, T lymphocytes, and basophils, which play a vital role in the recruitment of these leukocytes from the blood circulation to injured tissue, hence MCP-1 was described to be among the major markers implicated in the pathogenesis of several conditions associated with mononuclear cell infiltration. Previous reports indicated that decreased MCP-1 level reduces atherosclerosis [87]. Under normal conditions, another variant; CX3CR1^hi^ macrophages also produces IL-10, an anti-inflammatory cytokine marker known to maintain mucosal homeostasis [87]. In addition, other innate effector cells such as eosinophils are produced by Ly6C^hi^ monocytes through secretion of CCL11 (eotaxin). More importantly, Ly6C^hi^ monocytes are reported to directly regulate the pathogenic effects of neutrophils and the production of TNF-α and reactive oxygen species (ROS) by neutrophils in a PGE2-dependent manner [88]. Excessive accumulation of lipid-laden foam cells in endothelium signifies the earliest manifestation of an atherosclerotic lesion (Figure 5). These foam cells were reported to be derived from circulating monocytes after being adhered to the vascular endothelium. MCP-1 recruited-monocytes penetration of the vascular endothelium is associated with the responses to the gradient of chemotactic factors secreted from cells of the vascular endothelium wall. Basal NO inhibition by *N*^G^-nitro-l-arginine (L-NAG) upregulates endothelial MCP-1 mRNA expression and protein secretions. Alterations in MCP-1 mRNA expression and protein secretions are associated with the changes in chemotactic activities of cell-conditioned media for monocytes. High levels of MCP-1 mRNA have been investigated in many pathologic conditions, including atherosclerosis, rheumatoid arthritis, and glomerulonephritis [86]. Apart from monocyte recruitment, MCP-1 has been suggested to induce non-leukocytes to produce cytokines and adhesion molecules. MCP-1 also plays a vital role in the activation of inflammatory markers in vascular endothelium, including stimulated interleukin-6 (IL-6) secretion and intercellular adhesion molecule-1 (ICAM-1) synthesis [7]. MCP-1 induction of monocytes and neutrophils provides a significant contribution to coagulation by expression of other inflammatory markers and tissue factors, which are upregulated during inflammation [7]. Macrophages infiltration of the adipose tissues is the major contributor of inflammation associated with endothelial dysfunction [14]. Furthermore, activated neutrophils together with histone and other associated proteins are capable of expelling their DNA, forming an extracellular DNA termed as neutrophil extracellular traps (NETs), which is associated with antibacterial activities and induction of strong coagulatory responses [89]. The expression of TF is also stimulated both in vascular endothelial cells and monocytes by obesity-related cytokines [90]. Additionally, activated macrophages interact with adipocytes and preadipocytes to promote the production of inflammatory cytokines, thereby facilitating the inflammatory state in the vital organs, endothelial cells, and blood vessels leading to endothelial dysfunction [50]. The CX3CR1 macrophages also synthesizes increased amounts of a cytokine, including IL-12, and IL-23, as well as iNOS, which act to stimulate TLR to become proinflammatory-mediated cell [90,91].

## 5. Clinical Implication of Obesity-Induced Endothelial Dysfunction

Atherosclerosis describes a condition associated with the thickening of the arterial wall resulting from the massive deposition of LDL-cholesterol and triglycerides, leading to the development of multiple plaques within the blood vessels. Atherogenesis signified detrimental changes in the endothelial physiology progressing to atherosclerosis plaque with the subsequent progression to cardiovascular diseases [6]. Many contributing agents, including hyperlipidemia, hypertension, diabetes, elevated LDL- cholesterol levels, imbalance oxidative stress generated radicals, and adaptive smoking, as well as other disease factors described to be associated with vascular endothelial dysfunction were identified to function efficiently in the progression of atherosclerosis [6]. Generally, atherosclerosis is triggered by complex crosstalk within a different type of cells promoted by endothelial, which initiate the vicious cycle wherein the NF-κB stimulation results in oversecretion of adhesion molecules that bind to leukocytes, and subsequent increased production of inflammatory mediators that eventually activates the smooth muscle cells. These processes are characterized by vascular remodeling leading to atherosclerotic plaque and narrowing of the endothelial lumen [92,93,94]. Moreover, other agents such as shear stress can also promote endothelial cells dysfunction, specifically turbulent flow in the bloodstream. Generally, disturbed blood flow, including turbulent blood flow or oscillatory conditions causes shear stress and the subsequent over expression of proinflammatory genes associated with elevated endothelial cell layers permeability leading to atherosclerosis. However, unidirectional laminar blood flow has been known to inhibit endothelial activation and is attributed to resistance to atherosclerosis [95]. Molecular agents such as NF-kB, integrin, and matrix-dependent are thought to mediate blood flow-induced endothelial cell activation, while focal adhesion kinase was reported to regulate NF-κB phosphorylation and transcriptional activities associated with blood flow [16]. 

### Molecular Mechanisms Linked to the Progression of Atherosclerosis

Damage on the vascular endothelium caused by induced stress or any other agent leads to the expression of adhesion molecules and secretion cytokines and chemokines by the injured endothelium. Monocytes and other leucocytes from the blood are attracted to the site of injury by chemokines, specifically the MCP-1. Initially, monocytes are attached to the endothelial lumen through molecular interactions with adhesion molecules, they transmigrate to the subendothelium, differentiate, and mature to macrophages that release cytokines [6]. Once there are elevated levels of LDL and cholesterol, the LDL-c penetrates and infiltrates the subendothelium, and oxidizes to ox-LDL mediated by ROS and is retained in the intima. This results in the activation of endothelium leading to the transmigration and proliferation of leukocytes (macrophages and T-lymphocytes). The macrophages then take up accumulated ox- LDL-c forming cells and atherogenesis while T-lymphocytes differentiate to T-helper cells. These processes are associated with the secretion of proinflammatory cytokines, which combine with other growth factors to stimulate smooth muscle cells proliferation and migration into the sub-endothelial space (Figure 5). This indicates a fundamental stage in responding to vascular injury and the formation of a fibrous cap with the increased extracellular matrix, causing the thickening of the intima and the subsequent formation of atherosclerotic plaque [14]. Despite all these, endothelial dysfunction associated with atherosclerosis can be minimized through physical activities and changes in lifestyle for weight loss with minimal adipose tissue-related inflammation and increased NO availability [96]. 

## 6. Therapeutic Approaches Targeting Endothelial Dysfunction

The endothelium is an active layer lining the blood vessels characterized with regulated capacity for self-renewal and repair, through the terminal differentiating cells that regulate its proliferation potentials. Hence, the vascular endothelial damage–repair mechanisms are achieved through the contributions of endothelial progenitor cells (EPCs) in blood circulation [97]. Intravenous infusion of glucagon-like peptide-1 (GLP-1) continuously was reported to have significant improvement in vascular endothelial dysfunction. At molecular levels, GLP-1 receptors are synthesized in vascular endothelial cells, which functions to elevate the synthesis of NO with the subsequent inhibition of the endothelial cell adhesion factors expression [98]. Physical activities coupled with changes in lifestyle were identified to drastically decrease the inflammatory biochemical markers level associated with vascular endothelial dysfunction in obesity [99]. Diet-induced weight loss has the potential to reduce the levels of biomarkers of endothelial dysfunction and inflammation in overweight and obese patients with type 2 diabetes, independent of the medical chemotherapeutic procedure [60,82]. Drug treatment of fenofibrate has been reported to decrease the postprandial secretion of inflammatory cytokines and chemokines such as macrophages inflammatory protein-1α [5]. Paraoxonase-1 was proved to have both anti-inflammatory and antioxidative properties promoting the secretion of HDL-mediated eNOS, which inhibits myeloperoxidase inflammatory activity in the vascular endothelial cells [100]. Furthermore, 5-lipoxygenase inhibitors, 5-lipoxygenase, inhibitors of adhesion molecules, CCR2 and CCR5 inhibitors, and methotrexate-related drugs have been shown to protect vascular endothelium [61]. Other drug inhibitors such as cytokine IL-1β with canakinumab were adapted to control cardiovascular-related diseases, while the inhibitors of NLRP3 inflammasome with MCC950 were reported to reduce myocardial infarction size and functions. The endothelial glycocalyx has been identified as regulating vascular endothelial wall functions and integrity [100]. Several clinical trials are being conducted to examine whether anti-inflammatory treatment such as methotrexate therapy (TETHYS trial and CIRT trial) improves cardiovascular outcomes [60,82]. Additionally, randomized placebo-controlled, double-masked clinical trials of salsalate, IL1Ra, and anti-TNF-α are being adapted to investigate whether these anti-inflammatory approaches change disease risk in obesity, type 2 diabetes, and atherosclerotic cardiovascular disease [60]. Understanding the molecular basis of vascular endothelial dysfunction is needed to develop more strategies to at least reduce the incidence of cardiovascular-related diseases associated with obesity.

## 7. Conclusions

Vascular endothelial dysfunction serves as an initiator contributing extensively to the progression of obesity-related metabolic syndromes. Increased levels of circulating proinflammatory with the subsequent reduction of anti-inflammatory markers leading to vascular endothelial dysfunction were observed in obesity and its related metabolic syndromes. Thorough knowledge of the molecular mechanisms of obesity linked to endothelial dysfunction will assist in preventing and controlling obesity-induced inflammation. Finally, understanding the mechanisms employed by MCP-1 mediating immune system and vascular inflammation provide an opportunity for the prevention and controls of endothelial dysfunction in obesity. 

## Figures and Tables

**Figure 1 biomolecules-10-00291-f001:**
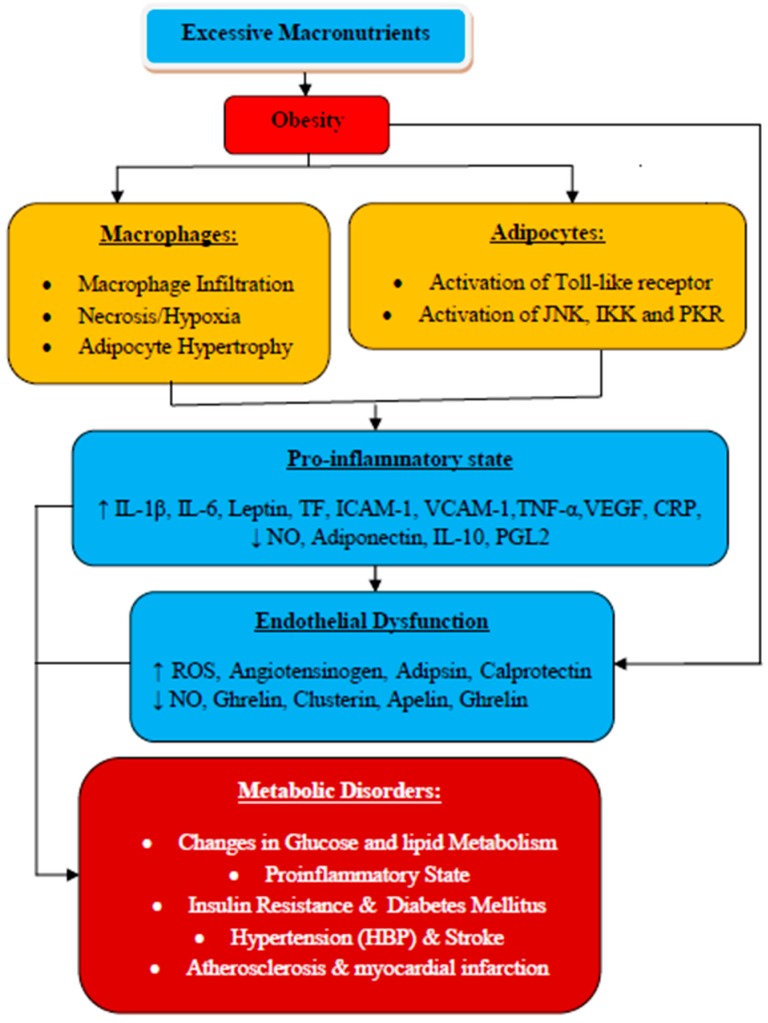
Obesity, inflammation, and related metabolic syndrome. The linking mechanisms: Obesity results from the accumulation of excessive macronutrients in the adipose tissues, which stimulate them to release inflammatory mediators such as TNF-α and IL-6, and the subsequent decreased production of adiponectin, leading to the predisposition of the endothelium to a proinflammatory state gearing to endothelial dysfunction. In addition, excessive accumulation of free fatty acids also activates proinflammatory serine kinase cascades, such as IkB kinase, Toll-like receptor, and c-Jun N-terminal kinase, which in turn facilitate adipose tissue to release IL-6 that triggers hepatocytes to produce and secrete CRP. All these are associated with the development of cardiovascular diseases, including atherosclerosis and other metabolic syndromes, as well as non-cardiovascular diseases such as renal diseases. A decreased level of adiponectin is also characterized by impaired fasting glucose, coronary artery calcification, and stroke. Hypoxia is described to be the etiology of necrosis and macrophage infiltration into adipose tissue, leading to overproduction of proinflammatory mediators [15].

**Figure 2 biomolecules-10-00291-f002:**
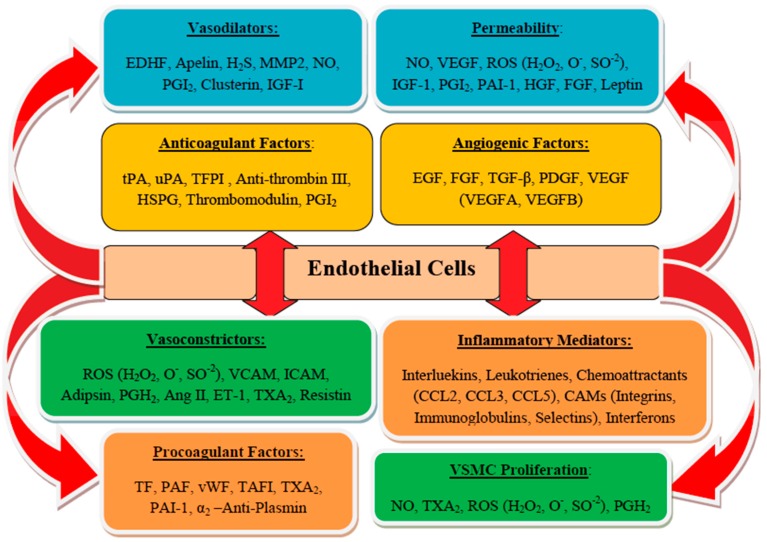
Important functions of endothelial cells: Vascular endothelium (VE) regulates vascular homeostasis through maintaining a delicate balance between the secretion of vasodilators and vasoconstrictors. It synthesizes a series of bioactive substances that moderate vascular tone, control permeability, regulate proliferation and migration of smooth muscle cells, decrease leucocyte migration, and regulate platelet adhesion and aggregation. VE also controls cellular adhesion, vascular inflammation, and angiogenesis [5].

**Figure 3 biomolecules-10-00291-f003:**
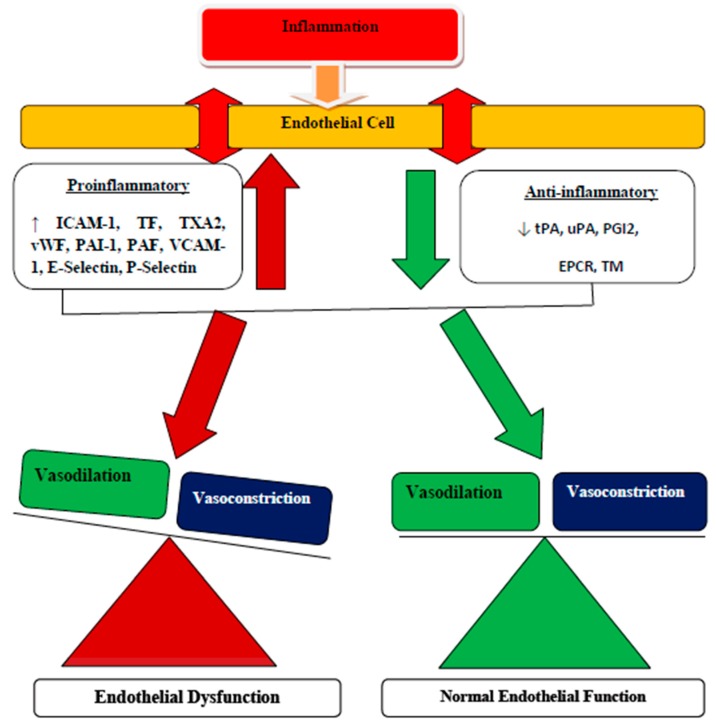
Mechanisms of endothelial cell (ECs) dysfunction associated with inflammation. Inflammation leads to an imbalance between proinflammatory and procoagulant, and anti-inflammatory and anticoagulant properties of the endothelium, thus contributing to disturbance of the hemostatic system. Activated or injured ECs mostly secrete procoagulant or antifibrinolytic components, resulting in the subsequent reduction in the expression of anticoagulant and profibrinolytic components. Additionally, activated ECs can express other factors such as TF and adhesion molecules, which have an important role in mediating the interaction of neutrophils and platelets with endothelium, further promoting the inflammatory and hemostatic responses. These disturbances shift the function of ECs from an anticoagulant, anti-inflammatory, and vasodilatory state to a proinflammatory and procoagulant state [49,50].

**Figure 4 biomolecules-10-00291-f004:**
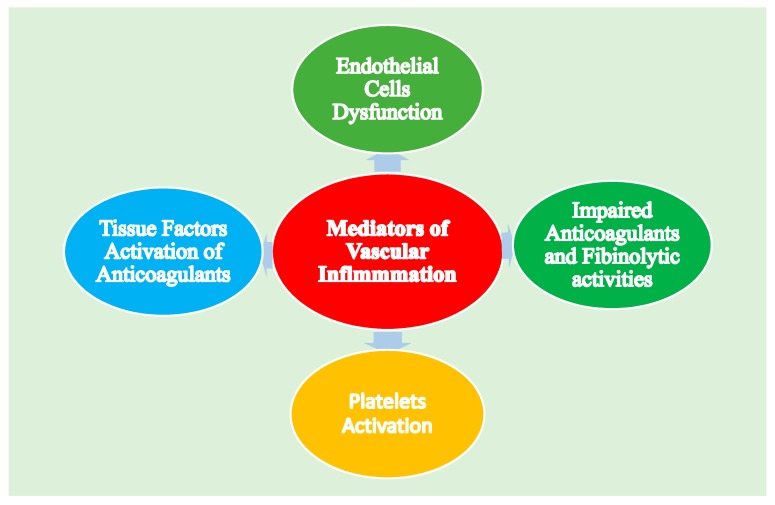
Common mediators of inflammation associated with the disturbance of the fibrinolytic system: The main mediators of inflammation-induced activation of the hemostatic system, particularly the proinflammatory cytokines, include tumor necrotic factor-α (TNF-α), interleukin-1 (IL-1), and interleukin-6 (IL-6), which trigger the disturbances of the hemostatic system in different mechanisms, including platelet activation, TF-mediated activation of coagulation cascade, impaired function of anticoagulant pathways and fibrinolytic activities, and endothelial cell dysfunction [50].

**Figure 5 biomolecules-10-00291-f005:**
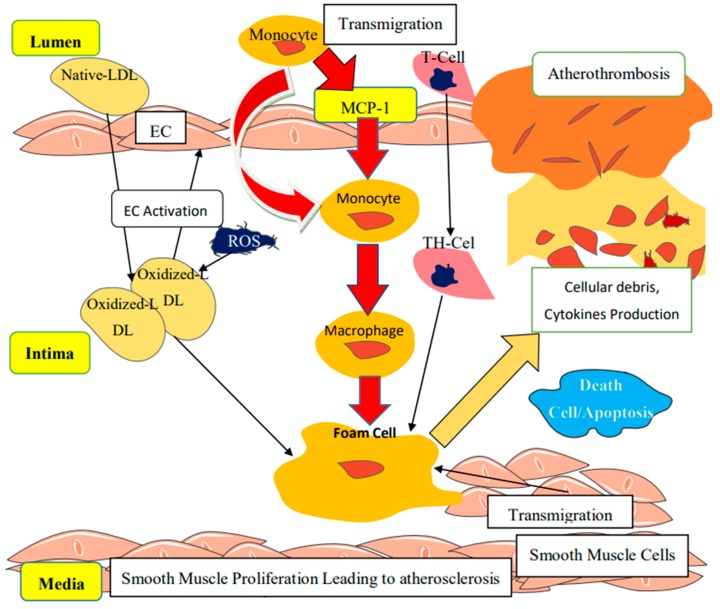
Monocyte chemoattractant protein-1 (MCP-1) contributions to endothelial dysfunction during atherosclerosis. Damage on the vascular endothelium leads to the secretion of cytokines and chemokines and the expression of adhesion molecules by the injured endothelium. Leucocytes from the blood are attracted to the site of injury by chemokines, specifically the MCP-1. Initially, monocytes are attached to the endothelial lumen through molecular interactions with adhesion molecules. They transmigrate to the subendothelium, differentiate, and mature to macrophages that release cytokines. Once there are elevated levels of low-density lipoprotein (LDL) and cholesterol, the LDL-c penetrates and infiltrates the subendothelium, and oxidized to ox-LDL mediated by reactive oxidative species (ROS) and is retained in the intima. This results in the activation of endothelium leading to the transmigrations and proliferation of leukocytes (macrophages and T-lymphocytes). The macrophages then take up accumulated ox- LDL-c resulting forming cells and atherogenesis while T-lymphocytes differentiate to T-helper cells. These processes are associated with the secretion of proinflammatory cytokines, which combine with other growth factors to stimulate smooth muscle cells proliferation and migration into the sub-endothelial space (Figure 5). This indicates a fundamental stage in responding to vascular injury and the formation of a fibrous cap with the increased extracellular matrix, causing the thickening of the intima with the subsequent formation of atherosclerotic plaque.

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
