# Peer review of "Endothelial Dysfunction in Obesity-Induced Inflammation: Molecular Mechanisms and Clinical Implications"

_biomolecules, 2020, doi:10.3390/biom10020291_

Round 1
Reviewer 1 Report
The paper by Ibrahim KK et al., discuss the molecular mechanism by which adipose tissue during obesity promotes endothelial dysfunction. The review is very well structured however there is minor concerns the authors should take into consideration.
In the introduction part, the paragraph starting in line 56 should be introduced. There no rational why suddenly the author decided to talk about endothelial dysfunction and obesity. Line 61, the author should insert a reference The clinical implications and therapeutic approaches part need to be further developed.Author Response
Response to Reviewer 1 Comments
The paper by Ibrahim KK et al., discuss the molecular mechanism by which adipose tissue during obesity promotes endothelial dysfunction. The review is very well structured however there are minor concerns the authors should take into consideration.
Point 1: In the introduction part, the paragraph starting in line 56 should be introduced. There no rationale why suddenly the author decided to talk about endothelial dysfunction and obesity.
Thank you very much, Sir, for sparing your busy time to go through my manuscript. Your professional comments are highly appreciated and noted.
Response 1: Endothelial dysfunction (ED) results from an imbalance in the production of vasodilatory agents such as nitric oxide (NO), endothelial-derived hyperpolarizing factors (EDHF), prostacyclin (PGI2) and vasoconstricting agents including prostaglandin (PGH2), endothelin-1 (ET-1) and angiotensin-II (Ang-II). Under normal physiology, the balanced release of contracting and endothelial-derived relaxing factors is maintained [6]. Alterations in this balance predispose the vascular endothelium towards pro-thrombotic and pro-atherogenic states resulting in platelet activation, leukocyte adherence, vasoconstriction, pro-oxidation, mitogenesis, vascular inflammation impaired coagulation, atherosclerosis and thrombosis with subsequent cardiovascular diseases [6]. During obesity and its related metabolic syndromes, this delicate balance is disrupted, and hence contributing to the development and further progression to vascular endothelial dysfunction [6,5].
Point 2: Line 61, the author should insert a reference
Response 2: [2] Hernández, H. R.; Hernández, L. E. S.; Ramírez, G. R.; Reyes, R. M. A. Obesity and Inflammation: Epidemiology, Risk Factors, and Markers of Inflammation. Int. J. Endocrinol., 2013, 2013, 11.
Point 3: The clinical implications and therapeutic approaches part need to be further developed.
Response 3: Diet-induced weight loss has the potential to reduce the levels of biomarkers of endothelial dysfunction and inflammation in overweight and obese patients with type 2 diabetes independent of the medical chemotherapeutic procedure [83,61]. Several clinical trials are going on to examine whether anti-inflammatory treatment such as methotrexate therapy (TETHYS trial and CIRT trial) improves cardiovascular outcomes [83,61]. Additionally, randomized placebo-controlled, double-masked clinical trials of salsalate, IL1Ra and anti-TNF-α are being adapted to investigate whether these anti-inflammatory approaches change disease risk in obesity, type 2 diabetes and atherosclerotic cardiovascular disease [61].
Reviewer 2 Report
The authors, Ibrahim et al, have prepared a review manuscript based on the pathophysiology of obesity-induced vascular complications associated with inflammation. Whilst this is undoubtedly an important and timely topic to review, there are several reviews in the field (many published in recent years) which review the mechanisms of obesity-induced vascular damage (PMID: 30844720; 30399375; 29167167; 28447436; 25001649; 24627704; 23448493; 16302012; 24924994; 20634940; 14624132). It is therefore unclear how the review actively contributes to the literature and/or provides new insight into the field. There is some indication that the MCP-1 angle of the review could be this new insight however significant consideration is needed by the authors to establish what is new or different about their review and expand on this in the review. This needs to be clearly stated. The figures also need to be geared towards MCP-1 if this is the case.
Other comments:
Reference formatting issues: Page 15, line 541 Figures 1,2,3,4 – Appropriate legend needed to describe the figure and mechanisms rather than just the abbreviations Figure 3 - this figure is rather basic and needs to be significantly improved. There is differing font text, the text is not aligned with the figures. Figure 4 – the text for some of the bubbles need to be adjusted In the introduction, line 42 (page 1), there is an overemphasis on MCP-1 in inflammation in obesity - this has not been definitively shown in the literature and some reference to the controversy needs to be made Page 2, section 2. There is a need to expand on the epidemiology and statistics of prevalence related to obesity and obesity-related diseases such as metabolic syndrome
Author Response
Response to Reviewer 2 Comments
The authors, Ibrahim et al, have prepared a review manuscript based on the pathophysiology of obesity-induced vascular complications associated with inflammation. Whilst this is undoubtedly an important and timely topic to review, there are several reviews in the field (many published in recent years) which review the mechanisms of obesity-induced vascular damage (PMID: 30844720; 30399375; 29167167; 28447436; 25001649; 24627704; 23448493; 16302012; 24924994; 20634940; 14624132). It is therefore unclear how the review actively contributes to the literature and/or provides new insight into the field. There is some indication that the MCP-1 angle of the review could be this new insight however significant consideration is needed by the authors to establish what is new or different about their review and expand on this in the review. This needs to be clearly stated. The figures also need to be geared towards MCP-1 if this is the case.
Other comments:
Reference formatting issues: Page 15, line 541 Figures 1,2,3,4 – Appropriate legend needed to describe the figure and mechanisms rather than just the abbreviations Figure 3 - this figure is rather basic and needs to be significantly improved. There is differing font text, the text is not aligned with the figures. Figure 4 – the text for some of the bubbles need to be adjusted In the introduction, line 42 (page 1), there is an overemphasis on MCP-1 in inflammation in obesity - this has not been definitively shown in the literature and some reference to the controversy needs to be made Page 2, section 2. There is a need to expand on the epidemiology and statistics of prevalence related to obesity and obesity-related diseases such as metabolic syndrome
Point 1: It is, therefore, unclear how the review actively contributes to the literature and/or provides new insight into the field. There is some indication that the MCP-1 angle of the review could be this new insight however significant consideration is needed by the authors to establish what is new or different about their review and expand on this in the review. This needs to be clearly stated. The figures also need to be geared towards MCP-1 if this is the case.
Thank you a lot for the professional comments made. All your comments are highly appreciated and noted. Meanwhile, when you go through all the figures used, there are a lot of factors or agents added compared to previous reviews. The citations made on each figure are mere citations, was not much drawn from the previous works but still, review writing should be a build-up from previous works. Well, I’m just a young Ph. D student trying to improve on my scientific writing. Surely, your comments are professionals; I must consider them and improve while I’m still learning from you professionals. Thank you very much.
Response 1: MCP-1 is the major chemoattractant for monocytes, T lymphocytes, and basophils, and plays a very vital role in the recruitment of these leukocytes from the blood circulation to injured tissue, hence, MCP-1 was described to be among the major markers implicated in the pathogenesis of several conditions associated with mononuclear cell infiltration. Previous reports indicated that decreased MCP-1 level reduces atherosclerosis [88]. Under normal condition, another variant; CX3CR1hi macrophages also produces IL-10, an anti-inflammatory cytokine marker known to maintain mucosal homeostasis [88]. In addition, other innate effector cells such as eosinophils are produced by Ly6Chi monocytes through secretion of CCL11 (eotaxin). More importantly, Ly6Chi monocytes are reported to directly regulate the pathogenic effects of neutrophils and the production of TNF-α and Reactive Oxygen Species (ROS) by neutrophils in a PGE2-dependent manner [89]. Recent studies indicated that CCR2 is also present in vascular endothelial cells. Activation of CCR2 by MCP-1was reported to be responsible for the renewal of the vascular endothelium following injury, angiogenesis and collateral formation. These processes may be essential during inflammatory lesion and tumours metastasis such as atheromatous plaques. Unfortunately, the detailed mechanism by which MCP-1 promotes angiogenesis is still under investigation [87]. Excessive accumulation of lipid-laden foam cells in endothelium signifies the earliest manifestation of an atherosclerotic lesion. These foam cells were reported to be derived from circulating monocytes after their adhered to the vascular endothelium. MCP-1 recruited-monocytes penetration of the vascular endothelium is associated with the responses to the gradient of chemotactic factors secreted from cells of the vascular endothelium wall. Basal NO inhibition by NG-nitro-l-arginine (L-NAG), upregulates endothelial MCP-1 mRNA expression and protein secretions. Alterations in MCP-1 mRNA expression and protein secretions are associated with the changes in chemotactic activities of cell-conditioned media for monocytes. High levels of MCP-1 mRNA have been investigated in many pathologic conditions including atherosclerosis, rheumatoid arthritis and glomerulonephritis [87].
Point 2: Reference formatting issues: Page 15, line 541 Figures 1,2,3,4 – Appropriate legend needed to describe the figure and mechanisms rather than just the abbreviations Figure 3 - this figure is rather basic and needs to be significantly improved.
Response 2: changed to [7]. Takada, Y.; Hisamatsu, T.; Kamada, N.;Takayama, T.; Kobayashi, T.; Chinen, H. This information is current as of May 29, 2019. https://doi.org/10.4049/jimmunol.0804012
Figure 1: Obesity results from the accumulation of excessive macronutrients in the adipose tissues, which stimulate them to release inflammatory mediators such as TNF-α and IL-6, with the subsequent decreased production of adiponectin leading to a predisposition of the endothelium to a pro-inflammatory state and endothelial dysfunction. In addition, excessive accumulation of free fatty acids also activates pro-inflammatory serine kinase cascades, such as IkB kinase, Toll-like receptor and c-Jun N-terminal kinase, which in turn facilitate adipose tissue to release IL-6 that triggers hepatocytes to produce and secrete CRP. All these are associated with the development of cardiovascular diseases including atherosclerosis and other metabolic syndromes as well as non-cardiovascular diseases such as renal diseases. A decreased level of adiponectin is also characterized by impaired fasting glucose, coronary artery calcification, and stroke. Hypoxia is described to be the aetiology of necrosis and macrophage infiltration into adipose tissue leading to overproduction of pro-inflammatory mediators (Ellul et al., 2014).
Figure 2: Vascular endothelium (VE) regulates vascular homeostasis through the maintaining of a delicate balance between the secretion of vasodilators and vasoconstrictors. It syntheses a series of bioactive substances that moderate vascular tone, control permeability, regulate proliferation and migration of smooth muscle cells, decrease leucocyte migration, and regulate platelet adhesion and aggregation. VE also controls cellular adhesion, vascular inflammation, and angiogenesis (Cristina et al., 2018)
Figure 3: Inflammation leads to an imbalance between proinflammatory and procoagulant and anti-inflammatory and anticoagulant properties of the endothelium thus contributing to disturbances of the haemostatic system. Activated or injured ECs mostly secrete procoagulant or antifibrinolytic components with the subsequent reduction in the expression of anticoagulant and profibrinolytic components. Additionally, activated ECs can express other factors such as TF and adhesion molecules which play an important role in mediating the interaction of neutrophils and platelets with endothelium, promoting the inflammatory and haemostatic responses. These disturbances shift the function of ECs from an anticoagulant, anti-inflammatory and vasodilatory state to a proinflammatory and procoagulant state Margetic, 2012).
Figure 4: The main mediators of inflammation-induced activation of the haemostatic system particularly the proinflammatory cytokines such as TNF-α, IL-1 and IL-6 trigger the disturbances of the haemostatic system in different mechanisms including platelet activation, TF- mediated activation of coagulation cascade, impaired function of anticoagulant pathways and fibrinolytic activities, and endothelial cell dysfunction (Margetic, 2012)
Point 3: There is differing font text, the text is not aligned with the figures. Figure 4 – the text for some of the bubbles need to be adjusted
Response 3:
Figure 4: The main mediators of inflammation-induced activation of the haemostatic system particularly the proinflammatory cytokines such as TNF-α, IL-1 and IL-6 trigger the disturbances of the haemostatic system in different mechanisms including platelet activation, TF- mediated activation of coagulation cascade, impaired function of anticoagulant pathways and fibrinolytic activities, and endothelial cell dysfunction (Margetic, 2012).
Point 4: In the introduction, line 42 (page 1), there is an overemphasis on MCP-1 in inflammation in obesity - this has not been definitively shown in the literature and some reference to the controversy needs to be made Page 2, section 2.
Response 4: Adequate explanation of MCP-1 is given in response 1 above with some cited references included
Point 5: There is a need to expand on the epidemiology and statistics of prevalence related to obesity and obesity-related diseases such as metabolic syndrome.
Response 5:
Obesity is a worldwide problem affecting both developed ad poor resources countries [2]. It is defined as the excessive deposition of body fat, reported by the body mass index (BMI) and calculated as weight (kg) divided by height (meters) squared. Individuals with BMI values greater than19 and less than 25 kg/m2 are considered healthy while people with a BMI equal or greater than 25 kg/m2 are overweight and those with a BMI equal or greater than 30 kg/m2 are considered obese [2]. Although the prevalence of obesity in developed countries was reported to be slowed in the past few years, the rate in the developing countries continue to increase, and might have even triple in some developing countries over the past few years [3,4] This is strongly attributed to growing changes in lifestyle, reduced physical activity and availability of modified junk foods [4,5] Obesity was described to result from increased energy intake with subsequent reduced physical activities [2] It is a metabolic condition of chronic low-grade inflammation associated with high levels of inflammatory markers such as CRP, IL-6 and TNF-α. [2,6]. In this, the low-grade chronic inflammation is characterized by a variety of chronic diseases including cardiovascular disease, diabetes, hypertension, non-alcoholic fatty liver disease [2], hypercholesterolemia, asthma, arthritis, some cancers and general poor health condition and hence represents a significant burden on the global healthcare system [5]. The global prevalence of obesity increases the potential risk factors for developing chronic metabolic syndromes [7]. Significantly increased in the body weight is highly correlated with several metabolic complications such as cardiovascular disease, pulmonary complications, diabetes and some certain cancers [8,9] leading to diminished life expectancy [2,6]. Furthermore, overweight or obese pregnant women have increased risks of developing offspring with obesity and its other related metabolic complications in future life, which justifies a transgenerational cycle of obesity [5]
Reviewer 3 Report
Overall, Ibrahim and colleagues have nicely described Endothelial Dysfunction in obesity-induced inflammation. The review is nicely structure and fairly comprehensive. The english needs a major revise and I would suggest the services of a professional language writer/editor.
Points
The inclusion of epigenetic aging (accelerated through obesity) would be a nice inclusion. eg endothelial senesence. Lipid oxidation should also be discussed, and the effect of oxidation on the vascular system (eg. ROS and mitochondria) I'm concerned that the figures are all taken and modified from other papers. There is nothing new in them or additive to our current body of knowledge. Vascular calcification should also be discussed in this context (Obesity)Author Response
Response to Reviewer three Comments
Comments and Suggestions for Authors
Overall, Ibrahim and colleagues have nicely described Endothelial Dysfunction in obesity-induced inflammation. The review is nicely structured and fairly comprehensive. The English need a major revise and I would suggest the services of a professional language writer/editor.
Points
The inclusion of epigenetic ageing (accelerated through obesity) would be a nice inclusion. eg endothelial senescence. Lipid oxidation should also be discussed, and the effect of oxidation on the vascular system (eg. ROS and mitochondria) I'm concerned that the figures are all taken and modified from other papers. There is nothing new in them or additive to our current body of knowledge. Vascular calcification should also be discussed in this context (Obesity)
Point 1: The English need a major revise and I would suggest the services of a professional language writer/editor.
Response 1: Thank you a lot for the professional comments made. All your comments are noted and highly appreciated. So surprising, I quite agreed that I’m not very good in English but also, on average I should not be bad in English. These days, many factors (e.g Plagiarism checkers) are there to contribute significantly to incalculable errors in English writing or even poor English arrangement. Some of the Haematological jargons, scientific jargons or even some keywords in English cannot be replaced accurately. Tempering on some words can make a sentence (s) to sound somehow but still correct grammatically. Well, I’m still a young Ph. D student trying to improve on my scientific writing. Definitely, your comments are professionals, I must consider them and improve while still learning from you professionals. Thank you very much.
Point 2: The inclusion of epigenetic ageing (accelerated through obesity) would be a nice inclusion. eg endothelial senescence.
Response 2:
Endothelial Dysfunction during Vascular Aging and Cellular Senescence
Atherosclerosis leading to cardiovascular complications including myocardial infarction, stroke, and ischemic heart failure, is considered to be the main cause of death in the Western world [52]. Atherosclerosis is well known to be associated with diabetes, LDL, cholesterol, smoking and hypertension. Recent reports have indicated that ageing is also one of the important risk factors for atherosclerosis and continue as an independent contributor when all other known factors are excluded [52]. Atherosclerosis-induced endothelial dysfunction is therefore a disease of both organismal ageing and cellular senescence. During advanced age, ECs become flattened, enlarged with an increasingly polypoid nucleus and all features identified with cellular senescence [53]. These alterations are associated with angiogenesis, proliferation, cell migration and modulation in cytoskeleton integrity. Vascular endothelial Senescent show reduced endothelial NO production144 with the elevated endothelin-1 release, decreased expression of VCAM-1 and ICAM-1, raised activation of NF-kB, and enhanced vulnerability to apoptosis [53]. Thus, senescent EC is attributed with the loss of EC activities and a subsequent shift toward a proinflammatory and proapoptotic state, which are predicted to promote monocyte migration into the vessel wall leading to endothelial dysfunction [52]. Naturally, vascular endothelial ageing is associated with intimal and medial thickening with a gradual loss of vascular elasticity, resulting in vascular endothelial stiffness [54]. Natural and premature aged cells shared various common characteristic including alterations in cell proliferative potentials, changes in markers of cell senescence, increased apoptosis, increased DNA damage with tremendous telomere shortening and dysfunction, decreased medial VSMC, elevated collagen deposition, and fracture of the elastin lamellae leading to vessel dilation and expanded lumen size.6 These are promoted by age-associated with elevated glycated proteins, matrix metalloproteinase enzyme activities, and trophic stimuli including angiotensin II signalling, impair vascular endothelial elasticity progressing to hypertension [52]. Cell senescence is described as the irreversible loss of the ability of the cells to divide and has been categorized into; replicative senescence and stress-induced premature senescence (SIPS). While replicative senescence progresses with age and is identified by shortened telomeres at chromosomal ends, which then promotes a DNA damage response (DDR), the SIPS on the other hand is triggered by external stimuli, such as radiation and oxidizing agents, which trigger the intracellular senescence cascade prematurely resulting to vascular endothelial dysfunction [52]. Basically, both normal vascular ageing and atherosclerosis are associated with cellular senescence. Cellular senescence impairs cell proliferation resulting in irreversible growth arrest and impairs survival, due to an accumulation of nuclear and mitochondrial DNA damage, increased ROS, and a proinflammatory state. Vascular ageing and cellular senescence are also associated with increased expression of proinflammatory cytokines and adhesion molecules which further promote inflammation and affect the synthesis and maintenance of extracellular matrix proteins. Ageing can be identified by both structural changes and by many senescence-associated biomarkers [52].
Endothelial dysfunction and Epigenetic Modifications
Epigenetic modifications such as DNA methylation and histone acetylation are described as post-replication changes in chromatin without any alteration in the basal nucleotide code [52]. These mechanisms are essential in controlling gene activation and silencing and are connected to various age-related conditions; including atherosclerosis [55]. Epigenetic modifications are therefore utilized as biomarkers of cellular senescence in the vascular endothelial. Previous studies have recognized certain changes in the expression of methyltransferases indicating that alterations in methyltransferase expression are linked with hypomethylation of hypermethylated genomic regions that occur within the genes known to participate in lipid metabolism, proliferation and apoptosis. Atherogenic lipoproteins can induce DNA methylation and histone deacetylation [52]. These findings proposed a potential linked between epigenetic modification and atherosclerosis. Histone acetylation and deacetylation are known to contribute significantly to the progress of atherosclerosis during ageing through which inflammation, VSMC proliferation, and ECM composition can be modulated. Furthermore, other studies have indicated that mitochondrial gene p66 (Shc), a known longevity gene, promoter hypermethylation and histone acetylation that results in age-related enhancement of p66Shc production and its subsequent activation [56].
Point 3: Lipid oxidation should also be discussed, and the effect of oxidation on the vascular system (eg. ROS and mitochondria)
Response 3:
Endothelial Dysfunction Influenced by Reactive Oxygen Species (ROS) in Obesity
Oxidative stress described a state of an imbalance between the bodies pro-oxidant and antioxidant systems, leading to platelet aggregation, thrombus formation and subsequent endothelial dysfunction [57]. It also alters pancreatic insulin secretion and glucose metabolisms in muscle and adipose tissue. Obesity and its related metabolic abnormalities are associated with increased oxidative stress radicals with elevated expression of NEFAs, TNFa, CRP, IL-6, TNF-α and LDL cholesterol. ROS also serves as a precursor to Ox-LDL formation, essential to the progress of atherosclerotic lesions. Glucose auto-oxidation in hyperglycaemia and Protein glycation also contribute significantly to free radical formation [58]. In contrast, HDL particles have both anti-inflammatory and anti-oxidative activities attributed to its ability to inhibit obesity-associated dyslipidaemia. Other antioxidant defence mechanisms that are reduced in obesity include decreased erythrocyte glutathione and glutathione peroxides [59]. In adipose tissues, the secretions of functional adipocytes are tightly controlled by inflammatory and metabolic signals. Adiponectin and Leptin are prime hormones adiposity signals secreted from non-obese and non-inflamed adipocytes. While Leptin acts primarily in the hypothalamus to control food intake and energy consumption, adiponectin secretion, on the other hand, is related to a reduced total body fat mass, promoting the whole-body insulin sensitivity [59]. The secretion of other factors, including IL-6, IL-8, chemerin, MCP-1, PAI-1, RANTES, resistin, retinol-binding protein 4 (RBP4), TNF-α or visfatin is significantly raised in adipocytes [60]. Some of these factors induce peripheral complications, vascular endothelial cell dysfunction, atherosclerosis or cell-mediated inflammatory processes [61,62]. Adipocytes hypertrophy enhances Lipid peroxidation with the subsequent generation of ROS; particularly, the generation of reactive aldehyde species (4-hydroxyalkenals) from polyunsaturated fatty acids is elevated. The ω-3- and ω-6 PUFAs are the reactive aldehydes 4-hydroxy-2E-hexenal (4-HHE), 4-hydroxy- 2Enonenal (4-HNE) and 4-hydroxy-2E, 6Z-dodecadienal (4-HDDE) are the main peroxidation products from which 4-HNE has been identified [63]. This is generated from a series of non-enzymatic peroxidation reactions of 15-hydroperoxy- 5Z,8Z,11Z,3E-eicosatetraenoic (15-HpETE) and 13-hydroperoxy- 9Z,11E-octadecadienoic acid (13-HpODE), 15-lipoxygenase (15-LO)-mediated transformation of arachidonic acid (AA) and linoleic acid. Other compounds include the 12-hydroperoxy- 5Z,8Z,10E, 14-Zeicosatetraenoic acid (12-HpETE), the 12- lipoxygenase (12-LO) metabolite of AA, are also transformed to 4-HDDE. The non-enzymatic peroxidation of ω-3 PUFAs, such as α-linolenic acid, eicosatetraenoic acid and docosahexaenoic acid were also reported to generate 4-HHE [64]. ROS are generated from the cells as by-products of oxidative phosphorylation. ROS promote mitochondrial DNA damage and dysfunction by the addition of double bonds to or removing the hydrogen atom from DNA bases. ROS are increased by elevated levels of oxidized lipoproteins in atherosclerosis, with the most common form being reactive hydroxyl free radicals (OH). Oxidative DNA damage occurs in mitochondrial DNA and in both telomeric and nontelomeric regions [65]. Indeed, many free radicals such as hydrogen peroxide (H2O2), oxide radical (O-), superoxide radical (SO-), and the level of 8-oxo-deoxyguanosine (8-oxodG), an oxidized form of guanine contributing to oxidative DNA damage are elevated in VSMCs and macrophages [66]. Improved in antioxidant therapy targeted at reducing lipid peroxidation and neutralisation of these radicals may eliminate inflammation and improve adipose tissue functions in obesity and its other related metabolic syndrome[67]
Point 4: I'm concerned that the figures are all taken and modified from other papers. There is nothing new in them or additive to our current body of knowledge.
Response 4: It may be possible that some of the ideas are initiated from previous reviewers but when you go through all the figures used, there are a lot of factors or agents added compared to previous reviewers. The citations made on each figure is a mere citation, was not much drawn from the previous works but still, review writing should be a build-up from previous works. Well, I'm just a young Ph. D student trying to improve on my scientific writing. Definitely, your comments are professionals; I must consider them and improve while I'm still learning from you professionals. Thank you very much.
Point 5: Vascular calcification should also be discussed in this context (Obesity)
Response 5:
Vascular calcification (VC) is one of the mechanisms that influenced vascular remodelling due to dedifferentiation of vascular smooth muscle cells (VSMC), alterations in elastin, collagen and endothelial dysfunction [68]. VC increases the chances of cardiovascular mortality and morbidity, especially in individuals with obesity, type 2 diabetes mellitus (T2DM), and chronic kidney disease [69]. Previously, VC has been described as a degenerative disease or as a consequence of ageing [70]. However, recent studies have reported that the mechanism of the vascular calcification is a multifactorial tightly controlled process similar to osteochondrogenesis and is associated with inflammation, dysregulated metabolism, osteogenesis and advanced atherosclerosis [71]. The sclerotic process in the vascular endothelium is considered to start from the fatty streaks present as early as childhood, progressing to atherosclerotic lesions which would be found in many young adults, and may gradually advance to calcified lesions and plaques [72]. Several studies have identified the correlation between the ROS generation, particularly H2O2 and the progression of vascular calcification. An elevated level of ROS triggers MMP (matrix metalloproteinase) activity and alteration in collagen and elastin deposition [73]. MMPs are family of Zn2+- and Ca2+-dependent endopeptidases expressed by many different cell types. Vascular endothelial cells can also produce MMPs, which plays a very vital role in vascular calcification by degrading extracellular matrix (ECM) component [70]
Reviewer 4 Report
This is a fine review work on endothelial dysfunction which gives a broad and sound perspective of this condition in relation to obesity from the molecular and clinical perspective. There are at least 30 reviews on endothelial dysfunction published during this year in relation to several health conditions and only two related to obesity, however, none of them describes clinical and molecular mechanisms in such didactic manner in each of its paragraphs as this review does. There are two remarks however, one is that Figure 4 in its present version does not provide enlightenment to the information in the paragraph which calls the figure and second, it would have been an excellent addition to the work, if the authors had reviewed the state-of-the-art knowledge about the gut microbiota and endothelial dysfunction in relation to obesity.
Author Response
Response to Reviewer Four Comments
REVIEWER FOUR
Comments and Suggestions for Authors
This is a fine review work on endothelial dysfunction which gives a broad and sound perspective of this condition in relation to obesity from the molecular and clinical perspective. There are at least 30 reviews on endothelial dysfunction published during this year in relation to several health conditions and only two related to obesity, however, none of them describes clinical and molecular mechanisms in such didactic manner in each of its paragraphs as this review does. There are two remarks however, one is that Figure 4 in its present version does not provide enlightenment to the information in the paragraph which calls the figure and second, it would have been an excellent addition to the work, if the authors had reviewed the state-of-the-art knowledge about the gut microbiota and endothelial dysfunction in relation to obesity.
Point 1: There are two remarks however; one is that Figure 4 in its present version does not provide enlightenment to the information in the paragraph which calls the figure.
Response 1: Line 291 and new legend included
Vascular inflammatory mediators disturb the haemostatic system in various mechanisms such as endothelial cells dysfunctions, activation of platelets, and the TF-mediated stimulation of the plasma coagulation system, suppress functions of PAP and impaired fibrinolytic activities (Fig. 4).
Point 2: Second, it would have been an excellent addition to the work, if the authors had reviewed the state-of-the-art knowledge about the gut microbiota and endothelial dysfunction in relation to obesity
Response 2: Atherosclerosis is the most common risk factor for CVD, characterized by excessive deposition of cholesterol and recruitment of macrophages into vascular walls, which contributes to the development and formation of atherosclerotic plaques [74]. Gut microbiota is the collection of bacteria that inhabit in the gastrointestinal tract [75]. They are mainly composed of five phyla, namely Bacteroidetes, Firmicutes, Actinobacteria, Proteobacteria, and Cerrucomicrobia, in which Bacteroidetes and Firmicutes are found more abundant in obese individuals [74,75]. Several studies have indicated that gut microbiota plays a contributing role in atherosclerosis through modulating inflammation and the secretion of microbial metabolites [76]. Recent studies also have shown the influenced of gut dysbiosis to the development and progress of atherosclerosis and subsequently to the CVD [77]. Furthermore, the Akkermansia muciniphila was identified to promote gut barrier functions and have attenuating effects against atherosclerosis [78]. Additionally, the human gut microbiota is associated with obesity and some members of the gut microbiota found to be present in the faeces of atherosclerotic patients are also present in their plaques [79,80]. Human gut microbiota derives energy from dietary fibre through fermentation and produces short-chain fatty acids (SCFAs) to influence host lipid energy metabolism [81]. Diet plays a significant role in modulating microbial diversity and reports have indicated that the high-fat diet are associated with obesity, whereas fibre-rich diet has the potential for reducing the risks to obesity [82,83]. Gut microbiota play a critical role in haemostasis for maintaining human health, with gut dysbiosis contributing to the development and progress of various diseases including CVD, obesity, T2DM, NAFLD, and even some types of cancer [84]
Round 2
Reviewer 2 Report
The manuscript is significantly improved. Unfortunately it is still not clear how the review actively contributes to the literature and/or provides new insight into the field. Whilst there is a better indication that MCP1 is of new interest this should be more clearly outlined.
Author Response
Reviewer Two Comments:
The manuscript is significantly improved. Unfortunately, it is still not clear how the review actively contributes to the literature and/or provides new insight into the field. Whilst there is a better indication that MCP1 is of new interest this should be more clearly outlined.
Response to Reviewer Two Comments
Point 1: Unfortunately it is still not clear how the review actively contributes to the literature and/or provides new insight into the field.
Thank you a lot for the professional comments made. All your comments are noted and highly appreciated. Definitely, your comments are professionals, I must consider them, improve and deal with them appropriately while still learning from you professionals. Thank you very much for your wonderful comments. I have learned a lot from you.
Response 1: Many factors/agents that are specifically related to endothelial dysfunction in obesity are narrowed, extracted, sorted and included also new factors/agents in each figure. The figures are also drawn quite different from the previous figures. But still, review writing should be a build-up from previous works, I have improved the manuscript. Thanks a lot, Sir.
Point 2: Whilst there is a better indication that MCP1 is of new interest this should be more clearly outlined.
Response 2: The functions and contributions of MCP-1 to endothelial dysfunction in obesity was highlighted in detail. The mechanisms linking obesity, inflammation and endothelial dysfunction were also discussed.
Fig. 5: MCP-1 Contributions to Endothelial Dysfunction during Atherosclerosis: Damaged on the vascular endothelium leads to the secretion of cytokines and chemokines, and the expression of adhesion molecules by the injured endothelium. Leucocytes from the blood are attracted to the site of injury by chemokines, specifically the MCP-1. Initially, Monocytes are attached to the endothelial lumen through molecular interactions with adhesion molecules. They transmigrate to the subendothelium, differentiate, and mature to macrophages that release cytokines. Once there are elevated levels of LDL and cholesterol, the LDL-c penetrates and infiltrates the subendothelium, and oxidized to ox-LDL mediated by ROS and is retained in the intima. This results in the activation of endothelium leading to the transmigrations and proliferation of leukocytes (Macrophages and T-lymphocytes). The Macrophages then take up accumulated ox- LDL-c resulting to form cells and atherogenesis while T-lymphocytes differentiate to T-helper cells. These processes are associated with the secretions of proinflammatory cytokines, which combine with other growth factors to stimulate smooth muscle cells proliferation and migration into the sub-endothelial space (Fig. 5). This indicates a fundamental stage in responding to vascular injury and the formation of a fibrous cap with the increased extracellular matrix, causing the thickening of the intima with the subsequent formation of atherosclerotic plaque.

Reviewer 3 Report
Small proof editing required. The addition of the additional topics add to the review and make it more comprehensive.
Author Response
Reviewer Three Comments:
Small proof editing required. The addition of the additional topics add to the review and make it more comprehensive.
Response to Reviewer three Comments
Point 1: Small proof editing required
Thank you a lot for the professional comments made. All your comments are noted and highly appreciated. Definitely, your comments are professionals, I must consider them, improve and deal with them appropriately while still learning from you professionals. Thank you very much for your wonderful comments. I have learned a lot from you.
Response 1: The manuscript has undergone thorough editing and re-editing, hoping that I have convinced you a little. Thanks a lot, Sir.